# Non-Monotonic Effects of Width on Seed-to-Seed Functional Similarity in Finite Neural Networks

## Abstract

Random seed sensitivity is a ubiquitous yet poorly characterized aspect of neural network training. Motivated by infinite-width intuitions suggesting increased determinism with model size, we investigate how network width affects seed-to-seed *functional* similarity in practical finite networks. Using a controlled sweep over multilayer perceptrons trained on MNIST with fixed depth, optimization, and data subsets, we measure pairwise similarity between independently trained models via output-logit correlation and prediction agreement. We find that functional similarity is *non-monotonic* in width: it increases at small widths, peaks at intermediate widths, and decreases substantially at larger widths, while validation-loss variance across seeds increases. These results suggest that increased capacity can expand the diversity of reachable low-loss solutions in finite regimes, cautioning against naive monotonic stabilization assumptions.

## 1 Introduction

Training neural networks with identical architectures and hyperparameters but different random seeds frequently yields different trained models. This variability impacts reproducibility and the reliability of empirical conclusions. A common intuition,partly motivated by infinite-width analyses such as the Neural Tangent Kernel (NTK) framework, is that increasing width should reduce sensitivity to randomness by making training dynamics more predictable (Jacot et al., 2018).

At the same time, empirical studies of loss landscapes indicate that neural networks admit many distinct low-loss solutions and complex geometry (Garipov et al., 2018; Draxler et al., 2018). These observations suggest that increased capacity may enable a broader set of reachable solutions. Despite these insights, it remains unclear how seed-to-seed *functional* variability depends on width in finite, practical networks.

In this work, we empirically characterize how width affects seed-to-seed functional similarity under a controlled training protocol. We fix depth, dataset, optimizer, and training schedule, vary only width and random seed, and quantify output-level similarity between trained models. Our primary finding is a clear non-monotonic relationship: similarity peaks at intermediate widths and decreases at larger widths, while validation-loss variance increases.

## 2 Related Work

**Infinite-width limits.** NTK theory formalizes that, under suitable conditions, wide neural networks admit training dynamics that converge to kernel regression in the infinite-width limit (Jacot et al., 2018). While these results motivate a notion of increasing predictability with width, they do not directly describe finite-width behavior.

**Loss landscapes and multiplicity of solutions.** Work on mode connectivity and connected low-loss paths highlights that neural networks can admit many solutions with comparable loss values (Garipov et al., 2018; Draxler et al., 2018). Such multiplicity may become more pronounced with increasing capacity.

Table 1: Experimental configuration (fixed across all runs).

| Component | Setting |
|---|---|
| Dataset | MNIST |
| Train subset | 20,000 fixed samples |
| Eval subset | 5,000 fixed samples |
| Model | 4-layer MLP (3 hidden + output), ReLU |
| Widths | $\{64, 128, 256, 512, 1024, 2048\}$ |
| Optimizer | Adam |
| Learning rate | $10^{-3}$ |
| Batch size | 128 |
| Epochs | 20 |
| Seeds per width | 20 |

## 3 EXPERIMENTAL SETUP

### 3.1 TASK AND DATASET

We use MNIST classification with standard preprocessing: `ToTensor` and normalization using mean 0.1307 and std 0.3081. For computational efficiency, we fix a subset of 20,000 images from the official training set for training, and a fixed subset of 5,000 images from the official test set for evaluation and similarity measurement. All experiments use the same fixed subsets.

### 3.2 MODEL

We use a fully connected MLP with depth 4: three hidden layers and one output layer. Hidden layers each have width $W$ and use ReLU activations. The input dimension is 784 (flattened MNIST) and the output dimension is 10.

### 3.3 OPTIMIZATION AND SWEEP

All models are trained for 20 epochs using Adam with learning rate $10^{-3}$, batch size 128, and cross-entropy loss. No batch normalization, dropout, or data augmentation is used.

We sweep widths:

$$W \in \{64, 128, 256, 512, 1024, 2048\},$$

and train 20 independent runs per width with seeds $\{0, \ldots, 19\}$, controlling initialization and data shuffling.

## 4 METRICS

For each width, we compare all $\binom{20}{2}$ pairs of trained models on the fixed evaluation set.

**Output correlation.** For a model $f$, let $z_f(x) \in \mathbb{R}^{10}$ denote its logits. We compute logits over the evaluation set, flatten all logits into a single vector, and compute the Pearson correlation between vectors for each model pair.

**Prediction agreement.** For each evaluation sample, we compute predicted labels via $\arg\max$ of logits and measure the fraction of samples on which two models predict the same class. We report mean pairwise agreement.

**Outcome variability.** We compute the variance of final validation loss across seeds for each width.

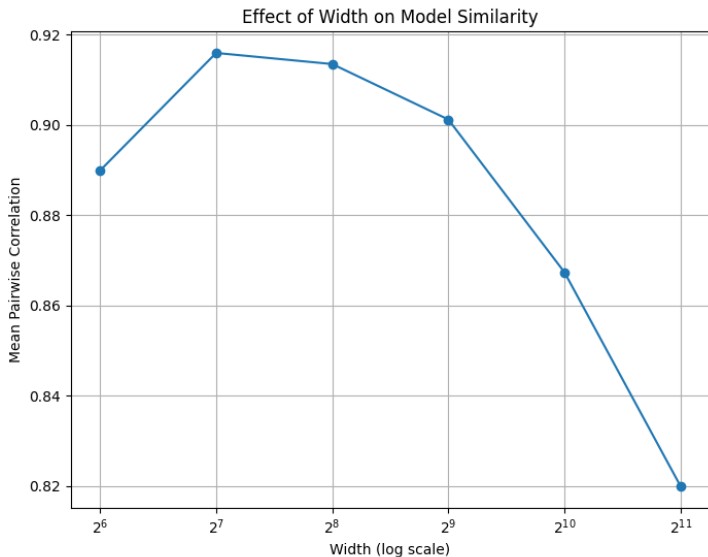

Figure 1: Mean pairwise output correlation vs width. Similarity peaks at intermediate widths and decreases at larger widths.

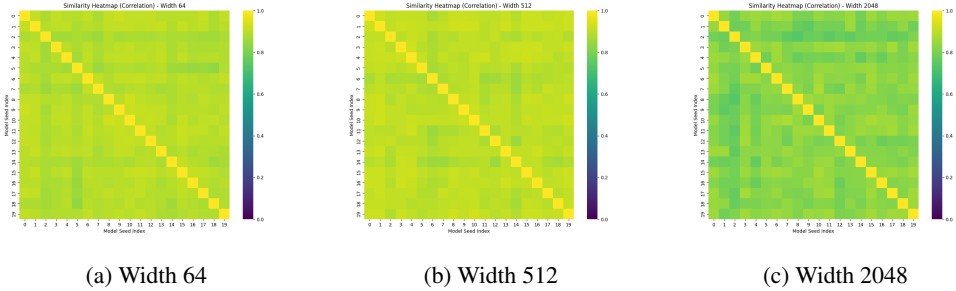

(a) Width 64          (b) Width 512          (c) Width 2048

Figure 2: Pairwise output-correlation heatmaps for selected widths.

## 5 RESULTS

### 5.1 FUNCTIONAL SIMILARITY IS NON-MONOTONIC IN WIDTH

Figure 1 shows mean pairwise output correlation as a function of width. Similarity increases from width 64 to intermediate widths (128–256) and then decreases steadily at larger widths (1024–2048), indicating increased seed-to-seed functional diversity at higher capacity.

### 5.2 HEATMAPS SHOW DECREASED UNIFORMITY AT LARGE WIDTH

Figure 2 visualizes pairwise correlation matrices for selected widths. The large-width setting (2048) exhibits visibly reduced uniformity compared to smaller widths, consistent with the trend in Figure 1.

### 5.3 OUTCOME VARIABILITY INCREASES WITH WIDTH

Figure 3 reports the validation-loss variance across seeds for each width. Variance increases markedly with width, indicating that optimization outcomes become more variable at larger widths under the fixed protocol.

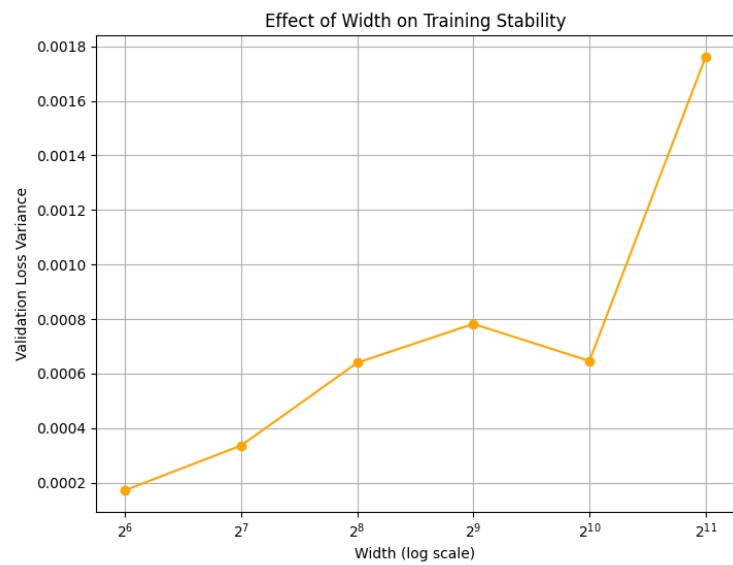

Figure 3: Variance of final validation loss across seeds vs width.

Table 2: Summary of key empirical findings (mean pairwise output correlation).

| Width $W$ | Mean output correlation |
|---|---|
| 64 | 0.8899 |
| 128 | 0.9159 |
| 256 | 0.9134 |
| 512 | 0.9012 |
| 1024 | 0.8673 |
| 2048 | 0.8200 |

## 6 DISCUSSION

Across a controlled width sweep, we observe that seed-to-seed functional similarity is not monotone in width. While similarity increases from small to intermediate widths, it decreases substantially for larger widths, and validation-loss variance increases. One plausible interpretation is that increasing capacity expands the set of reachable low-loss solutions, allowing different seeds to converge to more diverse functions. This observation aligns with the broader view that deep networks admit many solutions of comparable loss (Garipov et al., 2018; Draxler et al., 2018), and highlights that finite-width practice can differ qualitatively from infinite-width intuitions (Jacot et al., 2018).

## 7 CONCLUSION

We empirically characterize how width affects seed-to-seed functional similarity in finite neural networks under a controlled protocol. We find a clear non-monotonic relationship: similarity peaks at intermediate widths and decreases at larger widths, while validation-loss variance increases. These results caution against assuming monotonic stabilization with width and motivate further experimental study of training variability.

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
