# OpenReview forum: "Non-Monotonic Effects of Width on Seed-to-Seed Functional Similarity in Finite Neural Networks"
_ICLR.cc/2026/Workshop/Sci4DL — Submitted to Sci4DL 2026_

### Official Review · Reviewer_N1mL · 2026-02-17

**Fit:** 2
**Significance:** 1
**Confidence:** 3

**Summary:**

This paper examines how the output behavior of model functions vary across different random initializations for networks of different size (width).
The mean pairwise correlation between full logit vectors of different networks is used as the primary metric of function similarity.
The comparison is limited to 3 hidden layer MLPs trained on MNIST with 20 different random initialization seeds and 6 different widths.

The authors find that, somewhat surprisingly, the models' output variability across seeds first decreases as a function of width, before again increasing as the width is further increased.

**Strengths:**

The paper considers an interesting question that could assist in reconciling two differing views of function similarity for larger (but finite) networks: the stable infinite-width limit training behavior assumed by the NTK and the competing view of a larger hypothesis space of similarly generalizing solutions leading to greater possible variability.
Furthermore, the paper is also easy to follow and the experimental setup is clear.

**Suggestions:**

Unfortunately I find that the paper appears rushed and that the provided empirical evidence is still too limited at this point for presentation. My concerns/suggestions are as follows:

* It is difficult to draw any strong conclusions when only considering a single depth architecture on MNIST. I'd recommend repeating the analysis for other architectures and tiny-image datasets (FashionMNIST, CIFAR10, etc.).
* It would be useful to consider smaller models as well. The current evidence for "functional similarity first increases" is limited to the observed difference between width=64 and width=128 in Figure 1, and this trend is not reflected in the alternative metric of validation loss variance in Figure 3.
* Include training performance in the appendix. It is unclear if each model interpolates the training data. Perhaps some models simply require more training epochs to fit the same data per seed. If different samples are fit by each model it would explain the observed output variability.
* As far as I can tell, the "Prediction agreement" metric from Section 4 is never used. Figures 1 and 2 appear to use "output correlation", while Figure 3 uses "Outcome variability".
* Tables 1 and 2 are never referenced in the text.

---

### Official Review · Reviewer_1HJp · 2026-02-24

**Fit:** 1
**Significance:** 1
**Confidence:** 3

**Summary:**

The paper explores how width impacts the functional representation of a neural network at the end of training. In this setting, the authors train a 4-layer MLP with RelU activations on the MNIST dataset for 20 epochs under constant hyperparameters while varying the width from 64 to 2028, training 20 models at each width evaluated. The authors report that they find no monotonically increasing functional similarity relationship with increased width and instead observe that functional similarity peaks at intermediate widths of 128 and 256 and then continues to reduce at higher widths of 2048. This analysis is conducted primarily through prediction similarity, which is visualised through correlation heat maps and plots. While the results do provide some insight, due to the highly constrained and small-scale experimental setup, it is unclear how this work will be of interest to the broader community.

**Strengths:**

1. The paper sweeps over a range of width values, which helps to analyse multiple widths.
2. The setup for the experiment, while extremely small, is chosen well and kept consistent, which is appreciated.
3. The use of prediction agreement and heat maps is useful for visualising and quantifying agreement.
4. The findings are clear and would be interesting if explored for generality across architects.

**Suggestions:**

1. The paper states that only a subset of is used for computational efficiency; computation is not typically an issue when training small networks on MNIST. Furthermore, there is no information provided on how exactly the subset is obtained, which impacts reproducibility.
2. There is no justification as to why 20 seeds is an appropriate sample size for this study, given that training MLP models on MNIST is computationally cheap and therefore should leverage increased sample sizes, not just thirty per width. This would allow the use of statistical tests.
3. In Figures 1 and 3 there is no clear rationale as to why the x axis is in log scale when it could be just be the width values, also it is unclear how the values of 2^(6), 2^(7), 2^(8), 2^(9), 2^(10) and  2^(11) correspond to widths {64, 128, 256, 512, 1024, 2048}.
5. The paper states that "it remains unclear how seed-to-seed functional variability depends on width in finite, practical networks." however the experimental setup is very small (MLP and MNIST) and as a result there is no guarantee that such findings will scale to other more practical architectures and datasets so the claims in the paper are too strong for the experiments conducted. As a result, this means that the discussion from the results will be extremely limited for the workshop, as the evidence does not fully justify the claims made.
6. There is no standard deviation or standard error of the mean report for Figures 1 and 3, which reduces the ability to concretely say if there is indeed a meaningful difference in function across widths.
7. Just plotting the variance of the loss for Figure 3 does not provide the fullest picture and should be reported with the actual loss values so that convergence and similarity can be compared.

The paper claims that "Random seed sensitivity is a ubiquitous yet poorly characterised aspect of neural network training." While this is somewhat correct, the paper has not sufficiently referenced literature that studies seed sensitivity and functional outputs and therefore does not situate itself well with existing literature.

**Overall, due to the limited scale of the results, the lacklustre presentation and the overall rushed sense of the paper, I would recommend a rejection for the workshop.**

---

### Official Review · Reviewer_bkcZ · 2026-02-26

**Fit:** 1
**Significance:** 1
**Confidence:** 3

**Summary:**

This work examines the variability in neural network outputs across different seeds and network widths, and concludes that functional similarity is a non-monotonic function of the width. Specifically, similarity first increases at small widths and then decreases at larger widths.

**Strengths:**

- I appreciate that the paper thinks about the rather interesting problem of variation in neural networks functional outputs when considering architectural changes.

**Suggestions:**

- The experimental method is extremely unclear and overly lax. I suggest that, at the very least, error bars are illustrated for a paper that is examining variation in behavior across seeds.
- For an empirical paper, the experimental setup is extremely limited. It is difficult to trust any empirical result that is illustrated only on MNIST when using MLPs of sufficiently large width and depth (which is definitely the case in these experiments).
- I would also suggest looking into the usage of other correlation metrics -- the Pearson correlation coefficient can lead to highly misleading results when values are packed very close to each other. Indeed, this is likely to happen when evaluations are done with very large width networks on a (consequently) very easy problem such as MNIST.

Overall, I believe that this paper needs much more work and consideration put into it before it can be a research-level paper.

---

### Meta-Review · Area_Chair_Wgcx · 2026-03-02

**Recommendation:** Reject

**Metareview:**

Based on the reviews, this paper needs a major revision. I propose reject.

---

### Decision · Program_Chairs · 2026-03-02

Reject